# Deep Defense: Training DNNs with Improved Adversarial Robustness

**Ziang Yan**[1*]    **Yiwen Guo**[2,1*]    **Changshui Zhang**[1]

[1]Institute for Artificial Intelligence, Tsinghua University (THUAI),
State Key Lab of Intelligent Technologies and Systems,
Beijing National Research Center for Information Science and Technology (BNRist),
Department of Automation,Tsinghua University, Beijing, China
[2] Intel Labs China
yza18@mails.tsinghua.edu.cn  yiwen.guo@intel.com  zcs@mail.tsinghua.edu.cn

## Abstract

Despite the efficacy on a variety of computer vision tasks, deep neural networks (DNNs) are vulnerable to adversarial attacks, limiting their applications in security-critical systems. Recent works have shown the possibility of generating imperceptibly perturbed image inputs (a.k.a., adversarial examples) to fool well-trained DNN classifiers into making arbitrary predictions. To address this problem, we propose a training recipe named "deep defense". Our core idea is to integrate an adversarial perturbation-based regularizer into the classification objective, such that the obtained models learn to resist potential attacks, directly and precisely. The whole optimization problem is solved just like training a recursive network. Experimental results demonstrate that our method outperforms training with adversarial/Parseval regularizations by large margins on various datasets (including MNIST, CIFAR-10 and ImageNet) and different DNN architectures. Code and models for reproducing our results are available at https://github.com/ZiangYan/deepdefense.pytorch.

## 1   Introduction

Although deep neural networks (DNNs) have advanced the state-of-the-art of many challenging computer vision tasks, they are vulnerable to *adversarial examples* [34] (i.e., generated images which seem perceptually similar to the real ones but are intentionally formed to fool learning models).

A general way of synthesizing the adversarial examples is to apply worst-case perturbations to real images [34, 8, 26, 3]. With proper strategies, the required perturbations for fooling a DNN model can be $1000\times$ smaller in magnitude when compared with the real images, making them imperceptible to human beings. It has been reported that even the state-of-the-art DNN solutions have been fooled to misclassify such examples with high confidence [18]. Worse, the adversarial perturbation can transfer across different images and network architectures [25]. Such transferability also allows black-box attacks, which means the adversary may succeed without having any knowledge about the model architecture or parameters [28].

Though intriguing, such property of DNNs can lead to potential issues in real-world applications like self-driving cars and paying with your face systems. Unlike certain instability against random noise, which is theoretically and practically guaranteed to be less critical [7, 34], the vulnerability to adversarial perturbations is still severe in deep learning. Multiple attempts have been made to analyze and explain it so far [34, 8, 5, 14]. For example, Goodfellow et al. [8] argue that the main

reason why DNNs are vulnerable is their linear nature instead of nonlinearity and overfitting. Based on the explanation, they design an efficient $l_\infty$ induced perturbation and further propose to combine it with adversarial training [34] for regularization. Recently, Cisse et al. [5] investigate the Lipschitz constant of DNN-based classifiers and propose Parseval training. However, similar to some previous and contemporary methods, approximations to the theoretically optimal constraint are required in practice, making the method less effective to resist very strong attacks.

In this paper, we introduce "deep defense", an adversarial regularization method to train DNNs with improved robustness. Unlike many existing and contemporaneous methods which make approximations and optimize possibly untight bounds, we precisely integrate a perturbation-based regularizer into the classification objective. This endows DNN models with an ability of directly learning from attacks and further resisting them, in a principled way. Specifically, we penalize the norm of adversarial perturbations, by encouraging relatively large values for the correctly classified samples and possibly small values for those misclassified ones. As a regularizer, it is jointly optimized with the original learning objective and the whole problem is efficiently solved through being considered as training a recursive-flavoured network. Extensive experiments on MNIST, CIFAR-10 and ImageNet show that our method significantly improves the robustness of different DNNs under advanced adversarial attacks, in the meanwhile **no accuracy degradation is observed**.

The remainder of this paper is structured as follows. First, we briefly introduce and discuss representative methods for conducting adversarial attacks and defenses in Section 2. Then we elaborate on the motivation and basic ideas of our method in Section 3. Section 4 provides implementation details of our method and experimentally compares it with the state-of-the-arts, and finally, Section 5 draws the conclusions.

## 2  Related Work

**Adversarial Attacks.**  Starting from a common objective, many attack methods have been proposed. Szegedy et al. [34] propose to generate adversarial perturbations by minimizing a vector norm using box-constrained L-BFGS optimization. For better efficiency, Goodfellow et al. [8] develop the fast gradient sign (FGS) attack, by choosing the sign of gradient as the direction of perturbation since it is approximately optimal under a $\ell_\infty$ constraint. Later, Kurakin et al. [18] present an iterative version of the FGS attack by applying it multiple times with a small step size, and clipping pixel values on internal results. Similarly, Moosavi-Dezfooli et al. [26] propose DeepFool as an iterative $l_p$ attack. At each iteration, it linearizes the network and seeks the smallest perturbation to transform current images towards the linearized decision boundary. Some more detailed explanations of DeepFool can be found in Section 3.1. More recently, Carlini and Wagner [4] reformulate attacks as optimization instances that can be solved using stochastic gradient descent to generate more sophisticated adversarial examples. Based on the above methods, input- and network- agnostic adversarial examples can also be generated [25, 28].

**Defenses.**  Resisting adversarial attacks is challenging. It has been empirically studied that conventional regularization strategies such as dropout, weight decay and distorting training data (with random noise) do not really solve the problem [8]. Fine-tuning networks using adversarial examples, namely adversarial training [34], is a simple yet effective approach to perform defense and relieve the problem [8, 18], for which the examples can be generated either online [8] or offline [26]. Adversarial training works well on small datasets such as MNIST and CIFAR. Nevertheless, as Kurakin et al. [18] have reported, it may result in a decreased benign-set accuracy on large-scale datasets like ImageNet.

An alternative way of defending such attacks is to train a detector, to detect and reject adversarial examples. Metzen et al. [23] utilize a binary classifier which takes intermediate representations as input for detection, and Lu et al. [21] propose to invoke an RBF-SVM operating on discrete codes from late stage ReLUs. However, it is possible to perform attacks on the joint system if an adversary has access to the parameters of such a detector. Furthermore, it is still in doubt whether the adversarial examples are intrinsically different from the benign ones [3].

Another effective work is to exploit distillation [30], but it also slightly degrades the benign-set accuracy and may be broken by C&W's attack [4]. Alemi et al. [1] present an information theoretic method which helps on improving the resistance to adversarial attacks too. Some recent and contemporaneous works also propose to utilize gradient masking [29] as defenses [6, 35, 2].

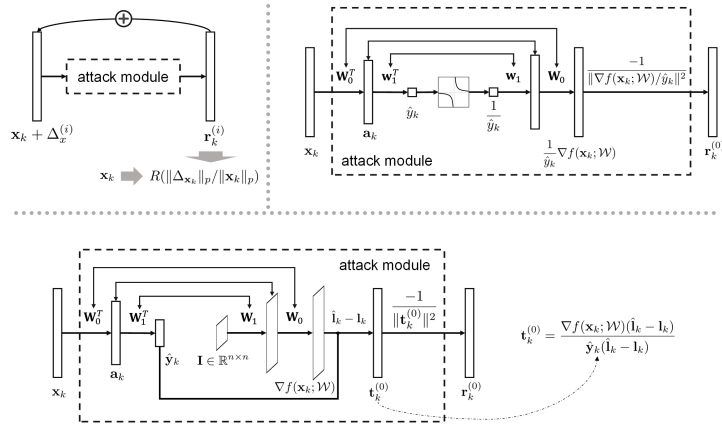

Figure 1: Top left: The recursive-flavoured network which takes a reshaped image $\mathbf{x}_k$ as input and sequentially compute each perturbation component by using a pre-designed attack module. Top right: an example for generating the first component, in which the three elbow double-arrow connectors indicate weight-sharing fully-connected layers and index-sharing between ReLU activation layers. Bottom: the attack module for $n$-class ($n \geq 2$) scenarios.

Several regularization-based methods have also been proposed. For example, Gu and Rigazio [9] propose to penalize the Frobenius norm of the Jacobian matrix in a layer-wise fashion. Recently, Cisse et al. [5] and Hein and Audriushchenko [14] theoretically show that the sensitivity to adversarial examples can be controlled by the Lipschitz constant of DNNs and propose Parseval training and cross-Lipschitz regularization, respectively. However, these methods usually require approximations, making them less effective to defend very strong and advanced adversarial attacks.

As a regularization-based method, our Deep Defense is orthogonal to the adversarial training, defense distillation and detecting then rejecting methods. It also differs from previous and contemporaneous regularization-based methods (e.g. [9, 5, 14, 31]) in a way that it endows DNNs the ability of directly learning from adversarial examples and precisely resisting them.

## 3 Our Deep Defense Method

Many methods regularize the learning objective of DNNs approximately, which may lead to a degraded prediction accuracy on the benign test sets or unsatisfactory robustness to advanced adversarial examples. We reckon it can be more beneficial to incorporate advanced attack modules into the learning process and learn to maximize a margin. In this section, we first briefly analyze a representative gradient-based attack and then introduce our solution to learn from it.

### 3.1 Generate Adversarial Examples

As discussed, a lot of efforts have been devoted to generating adversarial examples. Let us take the $l_2$ DeepFool as an example here. It is able to conduct 100% successful attacks on advanced networks. Mathematically, starting from a binary classifier $f : \mathbb{R}^m \to \mathbb{R}$ which makes predictions (to the class label) based on the sign of its outputs, DeepFool generates the adversarial perturbation $\Delta_\mathbf{x}$ for an arbitrary input vector $\mathbf{x} \in \mathbb{R}^m$ in a heuristic way. Concretely, $\Delta_\mathbf{x} = \mathbf{r}^{(0)} + ... + \mathbf{r}^{(u-1)}$, in which the $i$-th ($0 \leq i < u$) addend $\mathbf{r}^{(i)}$ is obtained by taking advantage of the Taylor's theorem and solving:

$$\min_\mathbf{r} \|\mathbf{r}\|_2 \quad \text{s.t.} \ f(\mathbf{x} + \Delta_\mathbf{x}^{(i)}) + \nabla f(\mathbf{x} + \Delta_\mathbf{x}^{(i)})^T \mathbf{r} = 0, \tag{1}$$

in which $\Delta_\mathbf{x}^{(i)} := \sum_{j=0}^{i-1} \mathbf{r}^{(j)}$, function $\nabla f$ denotes the gradient of $f$ w.r.t. its input, and operator $\| \cdot \|_2$ denotes the $l_2$ (i.e., Euclidean) norm. Obviously, Equation (1) has a closed-form solution as:

$$\mathbf{r}^{(i)} = -\frac{f(\mathbf{x} + \Delta_\mathbf{x}^{(i)})}{\|\nabla f(\mathbf{x} + \Delta_\mathbf{x}^{(i)})\|^2} \nabla f(\mathbf{x} + \Delta_x^{(i)}). \tag{2}$$

By sequentially calculating all the $\mathbf{r}^{(i)}$s with (2), DeepFool employs a faithful approximation to the $\Delta_{\mathbf{x}}$ of minimal $l_2$ norm. In general, the approximation algorithm converges in a reasonably small number of iterations even when $f$ is a non-linear function represented by a very deep neural network, making it both effective and efficient in practical usage. The for-loop for calculating $\mathbf{r}^{(i)}$s ends in advance if the attack goal $\mathrm{sgn}(f(\mathbf{x} + \Delta_{\mathbf{x}}^{(i)})) \neq \mathrm{sgn}(f(\mathbf{x}))$ is already reached at any iteration $i < u - 1$. Similarly, such strategy also works for the adversarial attacks to multi-class classifiers, which only additionally requires a specified target label in each iteration of the algorithm.

## 3.2  Perturbation-based Regularization

Our target is to improve the robustness of off-the-shelf networks without modifying their architectures, hence giving a $\|\Delta_{\mathbf{x}}\|_p$-based ($p \in [1, \infty)$) regularization to their original objective function seems to be a solution.

Considering the aforementioned attacks which utilize $\nabla f$ when generating the perturbation $\Delta_{\mathbf{x}}$ [34, 8, 26, 36], their strategy can be technically regarded as a function parameterized by the same set of learnable parameters as that of $f$. Therefore, it is possible that we jointly optimize the original network objective and a scaled $\|\Delta_{\mathbf{x}}\|_p$ as a regularization for some chosen norm operator $\|\cdot\|_p$, provided $\|\Delta_{\mathbf{x}}\|_p$ is differentiable. Specifically, given a set of training samples $\{(\mathbf{x}_k, \mathbf{y}_k)\}$ and a parameterized function $f$, we may want to optimize:

$$\min_{\mathcal{W}} \sum_k L(\mathbf{y}_k, f(\mathbf{x}_k; \mathcal{W})) + \lambda \sum_k R\left(-\frac{\|\Delta_{\mathbf{x}_k}\|_p}{\|\mathbf{x}_k\|_p}\right), \tag{3}$$

in which the set $\mathcal{W}$ exhaustively collects learnable parameters of $f$, and $\|\mathbf{x}_k\|_p$ is a normalization factor for $\|\Delta_{\mathbf{x}_k}\|_p$. As will be further detailed in Section 3.4, function $R$ should treat incorrectly and correctly classified samples differently, and it should be monotonically increasing on the latter such that it gives preference to those $f$s resisting small $\|\Delta_{\mathbf{x}_k}\|_p/\|\mathbf{x}_k\|_p$ anyway (e.g., $R(t) = \exp(t)$). Regarding the DNN representations, $\mathcal{W}$ may comprise the weight and bias of network connections, means and variances of batch normalization layers [16], and slops of the parameterized ReLU layers [12].

## 3.3  Network-based Formulation

As previously discussed, we re-formulate the adversarial perturbation as $\Delta_{\mathbf{x}_k} = g(\mathbf{x}_k; \mathcal{W})$, in which $g$ need to be differentiable except for maybe certain points, so that problem (3) can be solved using stochastic gradient descent following the chain rule. In order to make the computation more efficient and easily parallelized, an explicit formulation of $g$ or its gradient w.r.t $\mathcal{W}$ is required. Here we accomplish this task by representing $g$ as a "reverse" network to the original one. Taking a two-class multi-layer perceptron (MLP) as an example, we have $\mathcal{W} = \{\mathbf{W}_0, \mathbf{b}_0, \mathbf{w}_1, b_1\}$ and

$$f(\mathbf{x}_k; \mathcal{W}) = \mathbf{w}_1^T h(\mathbf{W}_0^T \mathbf{x}_k + \mathbf{b}_0) + b_1, \tag{4}$$

in which $h$ denotes the non-linear activation function and we choose $h(\mathbf{W}_0^T \mathbf{x}_k + \mathbf{b}_0) := \max(\mathbf{W}_0^T \mathbf{x}_k + \mathbf{b}_0, \mathbf{0})$ (i.e.as the ReLU activation function) in this paper since it is commonly used. Let us further denote $\mathbf{a}_k := h(\mathbf{W}_0^T \mathbf{x}_k + \mathbf{b}_0)$ and $\hat{y}_k := f(\mathbf{x}_k; \mathcal{W})$, then we have

$$\nabla f(\mathbf{x}_k; \mathcal{W}) = \mathbf{W}_0(\mathbf{1}_{>0}(\mathbf{a}_k) \otimes \mathbf{w}_1), \tag{5}$$

in which $\otimes$ indicates the element-wise product of two matrices, and $\mathbf{1}_{>0}$ is an element-wise indicator function that compares the entries of its input with zero.

We choose $\Delta_{\mathbf{x}_k}$ as the previously introduced DeepFool perturbation for simplicity of notation [1]. Based on Equation (2) and (5), we construct a recursive-flavoured regularizer network (as illustrated in the top left of Figure 1) to calculate $R(-\|\Delta_{\mathbf{x}_k}\|_p/\|\mathbf{x}_k\|_p)$. It takes image $\mathbf{x}_k$ as input and calculate each addend for $\Delta_{\mathbf{x}_k}$ by utilizing an incorporated multi-layer attack module (as illustrated in the top right of Figure 1). Apparently, the original three-layer MLP followed by a multiplicative inverse operator makes up the first half of the attack module and its "reverse" followed by a norm-based rescaling operator makes up the second half. It can be easily proved that the designed network is differentiable w.r.t. each element of $\mathcal{W}$, except for certain points. As sketched in the bottom of

Figure 1, such a network-based formulation can also be naturally generalized to regularize multi-class MLPs with more than one output neurons (i.e., $\hat{\mathbf{y}}_k \in \mathbb{R}^n$, $\nabla f(\mathbf{x}_k; \mathcal{W}) \in \mathbb{R}^{m \times n}$ and $n > 1$). We use $\mathbf{I} \in \mathbb{R}^{n \times n}$ to indicate the identity matrix, and $\hat{\mathbf{l}}_k$, $\mathbf{l}_k$ to indicate the one-hot encoding of current prediction label and a chosen label to fool in the first iteration, respectively.

Seeing that current winning DNNs are constructed as a stack of convolution, non-linear activation (e.g., ReLU, parameterized ReLU and sigmoid), normalization (e.g., local response normalization [17] and batch normalization), pooling and fully-connected layers, their $\nabla f$ functions, and thus the $g$ functions, should be differentiable almost everywhere. Consequently, feasible "reverse" layers can always be made available to these popular layer types. In addition to the above explored ones (i.e., ReLU and fully-connected layers), we also have deconvolution layers [27] which are reverse to the convolution layers, and unpooling layers [38] which are reverse to the pooling layers, etc.. Just note that some learning parameters and variables like filter banks and pooling indices should be shared among them.

### 3.4 Robustness and Accuracy

Problem (3) integrates an adversarial perturbation-based regularization into the classification objective, which should endow parameterized models with the ability of learning from adversarial attacks and resisting them. Additionally, it is also crucial not to diminish the inference accuracy on benign sets. Goodfellow et al. [8] have shown the possibility of fulfilling such expectation in a data augmentation manner. Here we explore more on our robust regularization to ensure it does not degrade benign-set accuracies either.

Most attacks treat all the input samples equally [34, 8, 26, 18], regardless of whether or not their predictions match the ground-truth labels. It makes sense when we aim to fool the networks, but not when we leverage the attack module to supervise training. Specifically, we might expect a decrease in $\|\Delta_{\mathbf{x}_k}\|_p / \|\mathbf{x}_k\|_p$ from any misclassified sample $\mathbf{x}_k$, especially when the network is to be "fooled" to classify it as its ground-truth. This seems different with the objective as formulated in (3), which appears to enlarge the adversarial perturbations for all training samples.

Moreover, we found it difficult to seek reasonable trade-offs between robustness and accuracy, if $R$ is a linear function (e.g., $R(z) = z$). In that case, the regularization term is dominated by some extremely "robust" samples, so the training samples with relatively small $\|\Delta_{\mathbf{x}_k}\|_p / \|\mathbf{x}_k\|_p$ are not fully optimized. This phenomenon can impose a negative impact on the classification objective $L$ and thus the inference accuracy. In fact, for those samples which are already "robust" enough, enlarging $\|\Delta_{\mathbf{x}_k}\|_p / \|\mathbf{x}_k\|_p$ is not really necessary. It is appropriate to penalize more on the currently correctly classified samples with abnormally small $\|\Delta_{\mathbf{x}_k}\|_p / \|\mathbf{x}_k\|_p$ values than those with relatively large ones (i.e., those already been considered "robust" in regard of $f$ and $\Delta_{\mathbf{x}_k}$).

To this end, we rewrite the second term in the objective function of Problem (3) as

$$\lambda \sum_{k \in \mathcal{T}} R\left(-c \frac{\|\Delta_{\mathbf{x}_k}\|_p}{\|\mathbf{x}_k\|_p}\right) + \lambda \sum_{k \in \mathcal{F}} R\left(d \frac{\|\Delta_{\mathbf{x}_k}\|_p}{\|\mathbf{x}_k\|_p}\right), \tag{6}$$

in which $\mathcal{F}$ is the index set of misclassified training samples, $\mathcal{T}$ is its complement, $c, d > 0$ are two scaling factors that balance the importance of different samples, and $R$ is chosen as the exponential function. With extremely small or large $c$, our method treats all the samples the same in $\mathcal{T}$, otherwise those with abnormally small $\|\Delta_{\mathbf{x}_k}\|_p / \|\mathbf{x}_k\|_p$ will be penalized more than the others.

## 4 Experimental Results

In this section, we evaluate the efficacy of our method on three different datasets: MNIST, CIFAR-10 and ImageNet [32]. We compare our method with adversarial training and Parseval training (also known as Parseval networks). Similar to previous works [26, 1], we choose to fine-tune from pre-trained models instead of training from scratch. Fine-tuning hyper-parameters can be found in the supplementary materials. All our experiments are conducted on an NVIDIA GTX 1080 GPU. Our main results are summarized in Table 1, where the fourth column demonstrates the inference accuracy of different models on benign test images, the fifth column compares the robustness of different models to DeepFool adversarial examples, and the subsequent columns compare the robustness to

Table 1: Test set performance of different defense methods. Column 4: prediction accuracies on benign examples. Column 5: $\rho_2$ values under the DeepFool attack. Column 6-8: prediction accuracies on the FGS adversarial examples.

| Dataset | Network | Method | Acc. | $\rho_2$ | Acc.@$0.2\epsilon_{\text{ref}}$ | Acc.@$0.5\epsilon_{\text{ref}}$ | Acc.@$1.0\epsilon_{\text{ref}}$ |
|---|---|---|---|---|---|---|---|
| MNIST | MLP | Reference | 98.31% | $1.11\times10^{-1}$ | 72.76% | 29.08% | 3.31% |
| | | Par. Train | 98.32% | $1.11\times10^{-1}$ | 77.44% | 28.95% | 2.96% |
| | | Adv. Train I | 98.49% | $1.62\times10^{-1}$ | 87.70% | 59.69% | 22.55% |
| | | Ours | **98.65%** | $\mathbf{2.25\times10^{-1}}$ | **95.04%** | **88.93%** | **50.00%** |
| | LeNet | Reference | 99.02% | $2.05\times10^{-1}$ | 90.95% | 53.88% | 19.75% |
| | | Par. Train | 99.10% | $2.03\times10^{-1}$ | 91.68% | 66.48% | 19.64% |
| | | Adv. Train I | 99.18% | $2.63\times10^{-1}$ | 95.20% | 74.82% | 41.40% |
| | | Ours | **99.34%** | $\mathbf{2.84\times10^{-1}}$ | **96.51%** | **88.93%** | **50.00%** |
| CIFAR-10 | ConvNet | Reference | 79.74% | $2.59\times10^{-2}$ | 61.62% | 37.84% | 23.85% |
| | | Par. Train | 80.48% | $3.42\times10^{-2}$ | 69.19% | 50.43% | 22.13% |
| | | Adv. Train I | 80.65% | $3.05\times10^{-2}$ | 65.16% | 45.03% | 35.53% |
| | | Ours | **81.70%** | $\mathbf{5.32\times10^{-2}}$ | **72.15%** | **59.02%** | **50.00%** |
| | NIN | Reference | 89.64% | $4.20\times10^{-2}$ | 75.61% | 49.22% | 33.56% |
| | | Par. Train | 88.20% | $4.33\times10^{-2}$ | 75.39% | 49.75% | 17.74% |
| | | Adv. Train I | 89.87% | $5.25\times10^{-2}$ | 78.87% | 58.85% | 45.90% |
| | | Ours | **89.96%** | $\mathbf{5.58\times10^{-2}}$ | **80.70%** | **70.73%** | **50.00%** |
| ImageNet | AlexNet | Reference | 56.91% | $2.98\times10^{-3}$ | 54.62% | 51.39% | 46.05% |
| | | Ours | **57.11%** | $\mathbf{4.54\times10^{-3}}$ | **55.79%** | **53.50%** | **50.00%** |
| | ResNet | Reference | 69.64% | $1.63\times10^{-3}$ | 63.39% | 54.45% | 41.70% |
| | | Ours | **69.66%** | $\mathbf{2.43\times10^{-3}}$ | **65.53%** | **59.46%** | **50.00%** |

FGS adversarial examples. The evaluation metrics will be carefully explained in Section 4.1. Some implementation details of the compared methods are shown as follows.

**Deep Defense.** There are three hyper-parameters in our method: $\lambda$, $c$ and $d$. As previously explained in Section 3.4, they balance the importance of the model robustness and benign-set accuracy. We fix $\lambda = 15, c = 25, d = 5$ for MNIST and CIFAR-10 major experiments (except for NIN, $c = 70$), and uniformly set $\lambda = 5, c = 500, d = 5$ for all ImageNet experiments. Practical impact of varying these hyper-parameters will be discussed in Section 4.2. The Euclidean norm is simply chosen for $\| \cdot \|_p$.

**Adversarial Training.** There exist many different versions of adversarial training [34, 8, 26, 18, 24, 22], partly because it can be combined with different attacks. Here we choose two of them, in accordance with the adversarial attacks to be tested, and try out to reach their optimal performance. First we evaluate the one introduced in the DeepFool paper [26], which utilizes a fixed adversarial training set generated by DeepFool, and summarize its performance in Table 1 (see "Adv. Train I"). We also test Goodfellow et al.'s adversarial training objective [8] (referred to as "Adv. Train II") and compare it with our method intensively (see supplementary materials), considering there exists trade-offs between accuracies on benign and adversarial examples. In particular, a combined method is also evaluated to testify our previous claim of orthogonality.

**Parseval Training.** Parseval training [5] improves the robustness of a DNN by controlling its global Lipschitz constant. Practically, a projection update is performed after each stochastic gradient descent iteration to ensure all weight matrices' Parseval tightness. Following the original paper, we uniformly sample 30% of columns to perform this update. We set the hyper-parameter $\beta = 0.0001$ for MNIST, and $\beta = 0.0003$ for CIFAR-10 after doing grid search.

## 4.1 Evaluation Metrics

This subsection explains some evaluation metrics adopted in our experiments. Different $l_p$ (e.g., $l_2$ and $l_\infty$) norms have been used to perform attacks. Here we conduct the famous FGS and DeepFool as representatives of $l_\infty$ and $l_2$ attacks and compare the robustness of obtained models using different defense methods. As suggested in the paper [26], we evaluate model robustness by calculating

$$\rho_2 := \frac{1}{|\mathcal{D}|} \sum_{k \in \mathcal{D}} \frac{\|\Delta_{\mathbf{x}_k}\|_2}{\|\mathbf{x}_k\|_2}, \tag{7}$$

in which $\mathcal{D}$ is the test set (for ImageNet we use its validation set), when DeepFool is used.

It is popular to evaluate the accuracy on a perturbed $\mathcal{D}$ as a metric for the FGS attack [9, 8, 5]. Likewise, we calculate the smallest $\epsilon$ such that 50% of the perturbed images are misclassified by our

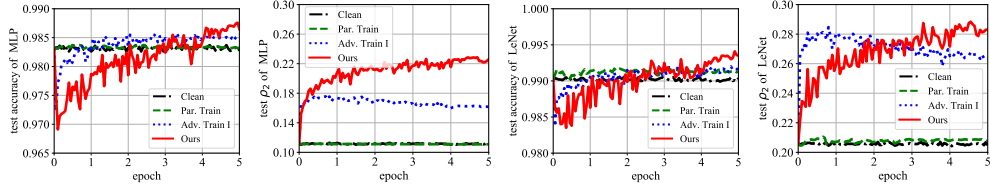

Figure 2: Convergence curves. From left to right: test accuracy and $\rho_2$ of MLP, and test accuracy and $\rho_2$ of LeNet. "Clean" indicates fine-tuning on benign examples. Best viewed in color.

regularized models and denote it as $\epsilon_{\text{ref}}$, then test prediction accuracies of those models produced by adversarial and Parseval training at this level of perturbation (abbreviated as "Acc.@$1.0\epsilon_{\text{ref}}$" in Table 1). Accuracies at lower levels of perturbations (a half and one fifth of $\epsilon_{\text{ref}}$) are also reported.

Many other metrics will be introduced and used for further comparisons in supplementary materials.

## 4.2 Exploratory Experiments on MNIST

As a popular dataset for conducting adversarial attacks [34, 8, 26], MNIST is a reasonable choice for us to get started. It consists of 70,000 grayscale images, in which 60,000 of them are used for training and the remaining are used for test. We train a four-layer MLP and download a LeNet [19] structured CNN model [2] as references (see supplementary materials for more details). For fair comparisons, we use identical fine-tuning policies and hyper-parameters for different defense methods We cut the learning rate by $2\times$ after four epochs of training because it can be beneficial for convergence.

**Robustness and accuracy.** The accuracy of different models (on the benign test sets) can be found in the fourth column of Table 1 and the robustness performance is compared in the last four columns. We see Deep Defense consistently and significantly outperforms competitive methods in the sense of both robustness and accuracy, even though our implementation of Adv. Train I achieves slightly better results than those reported in [26]. Using our method, we obtain an MLP model with **over $2\times$** better robustness to DeepFool and an absolute error decrease of 46.69% under the FGS attack considering $\epsilon = 1.0\epsilon_{\text{ref}}$, while the inference accuracy also increases a lot (from 98.31% to **98.65%** in comparison with the reference model. The second best is Adv. Train I, which achieves roughly $1.5\times$ and an absolute 19.24% improvement under the DeepFool and FGS attacks, respectively. Parseval training also yields models with improved robustness to the FGS attack, but they are still vulnerable to the DeepFool. The superiority of our method holds on LeNet, and the benign-set accuracy increases from 99.02% to **99.34%** with the help of our method.

Convergence curves of different methods are provided in Figure 2, in which the "Clean" curve indicates fine-tuning on the benign training set with the original learning objective. Our method optimizes more sophisticated objective than the other methods so it takes longer to finally converge. However, both its robustness and accuracy performance surpasses that of the reference models in only three epochs and keeps growing in the last two. Consistent with results reported in [26], we also observe growing accuracy and decreasing $\rho_2$ on Adv. Train I.

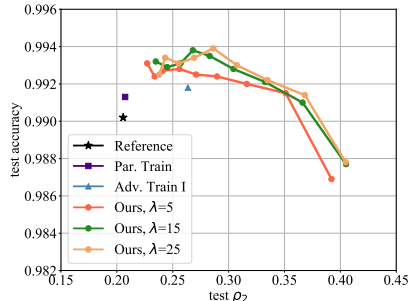

Figure 3: The performance of Deep Defense with varying hyper-parameters on LeNet. Best viewed in color.

In fact, the benefit of our method to test-set accuracy for benign examples is unsurprising. From a geometrical point of view, an accurate estimation of the optimal perturbation like our $\Delta_{\mathbf{x}_k}$ represents the distance from a benign example $\mathbf{x}_k$ to the decision boundary, so maximizing $\|\Delta_{\mathbf{x}_k}\|$ approximately maximizes the margin. According to some previous theoretical works [37, 33], such regularization to the margin should relieve the overfitting problem of complex learning models (including DNNs) and thus lead to better test-set performance on benign examples.

**Varying Hyper-parameters.** Figure 3 illustrates the impact of the hyper-parameters in our method. We fix $d = 5$ and try to vary $c$ and $\lambda$ in $\{5, 10, 15, 20, 25, 30, 35, 40, 45\}$ and $\{5, 15, 25\}$, respectively.

[2]https://github.com/LTS4/DeepFool/blob/master/MATLAB/resources/net.mat

Note that $d$ is fixed here because it has relatively minor effect on our fine-tuning process on MNIST. In the figure, different solid circles on the same curve indicate different values of $c$. From left to right, they are calculated with decreasing $c$, which means a larger $c$ encourages achieving a better accuracy but lower robustness. Conversely, setting a very small $c$ (e.g., $c = 5$) can yield models with high robustness but low accuracies. By adjusting $\lambda$, one changes the numerical range of the regularizer. A larger $\lambda$ makes the regularizer contributes more to the whole objective function.

**Layer-wise Regularization.** We also investigate the importance of different layers to the robustness of LeNet with our Deep Defense method. Specifically, we mask the gradient (by setting its elements to zero) of our adversarial regularizer w.r.t. the learning parameters (e.g., weights and biases) of all layers except one. By fixing $\lambda = 15$, $d = 5$ and varying $c$ in the set $\{5, 15, 25, 35, 45\}$, we obtain 20 different models. Figure 4 demonstrates the $\rho_2$ values and benign-set accuracies of these models. Different points on the same curve correspond to fine-tuning with different values of $c$ (decreasing from left to right). Legends indicate the gradient of which layer is not masked. Apparently, when only one layer is exploited to regularize the classification objective, optimizing "fc1" achieves the best performance. This is consistent with previous results that "fc1" is the most "redundant" layer of LeNet [11, 10].

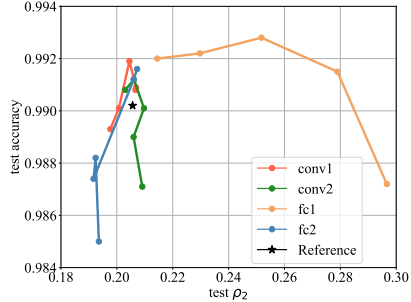

Figure 4: The performance of Deep Defense when only one layer is regularized for LeNet. Best viewed in color.

### 4.3 Image Classification Experiments

For image classification experiments, we testify the effectiveness of our method on several different benchmark networks on the CIFAR-10 and ImageNet datasets.

**CIFAR-10 results.** We train two CNNs on CIFAR-10: one with the same architecture as in [15], and the other with a network-in-network architecture [20]. Our training procedure is the same as in [26]. We still compare our Deep Defense with adversarial and Parseval training by fine-tuning from the references. Fine-tuning hyper-parameters are summarized in the supplementary materials. Likewise, we cut the learning rate by $2\times$ for the last 10 epochs.

Quantitative comparison results can be found in Table 1, in which the two chosen CNNs are referred to as "ConvNet" and "NIN", respectively. Obviously, our Deep Defense outperforms the other defense methods considerably in all test cases. When compared with the reference models, our regularized models achieve higher test-set accuracies on benign examples and gain absolute error decreases of 26.15% and 16.44% under the FGS attack. For the DeepFool attack which might be stronger, our method gains $2.1\times$ and $1.3\times$ better robustness on the two networks.

**ImageNet results.** As a challenging classification dataset, ImageNet consists of millions of high-resolution images [32]. To verify the efficacy and scalability of our method, we collect well-trained AlexNet [17] and ResNet-18 [13] model from the Caffe and PyTorch model zoo respectively, fine-tune them on the ILSVRC-2012 training set using our Deep Defense and test it on the validation set. After only 10 epochs of fine-tuning for AlexNet and 1 epoch for ResNet, we achieve roughly $1.5\times$ improved robustness to the DeepFool attack on both architectures, along with a slightly increased benign-set accuracy, highlighting the effectiveness of our method.

## 5   Conclusion

In this paper, we investigate the vulnerability of DNNs to adversarial examples and propose a novel method to address it, by incorporating an adversarial perturbation-based regularization into the classification objective. This shall endow DNNs with an ability of directly learning from attacks and precisely resisting them. We consider the joint optimization problem as learning a recursive-flavoured network to solve it efficiently. Extensive experiments on MNIST, CIFAR-10 and ImageNet have shown the effectiveness of our method. In particular, when combined with the FGS-based adversarial learning, our method achieves even better results on various benchmarks. Future works shall include explorations on resisting black-box attacks and attacks in the physical world.

**Acknowledgments**

This work is supported by NSFC (Grant No. 61876095, No. 61751308 and No.61473167) and Beijing Natural Science Foundation (Grant No. L172037).

## Footnotes

*The first two authors contributed equally to this work.

[1]Note that our method also naturally applies to some other gradient-based adversarial attacks.

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
