[Supplementary Material]

# Supplementary Material for Deep Defense: Training DNNs with Improved Adversarial Robustness

**Ziang Yan**[1*]    **Yiwen Guo**[2,1*]    **Changshui Zhang**[1]

[1]Institute for Artificial Intelligence, Tsinghua University (THUAI),
State Key Lab of Intelligent Technologies and Systems,
Beijing National Research Center for Information Science and Technology (BNRist),
Department of Automation,Tsinghua University, Beijing, China
[2] Intel Labs China
yza18@mails.tsinghua.edu.cn  yiwen.guo@intel.com  zcs@mail.tsinghua.edu.cn

## 1   More Evaluation Metrics and Attacks

Table 2: Test performance of different methods in the sense of: $l_2$ under DeepFool, $\rho_\infty$ under FGS, $l_2$ under the C&W attack, the prediction accuracy on PGD adversarial examples and the PASS score.

| Dataset | Network | Method | $l_2$ (DeepFool) | $\rho_\infty$ (FGS) | $l_2$ (C&W) | Acc. (PGD) | PASS |
|---|---|---|---|---|---|---|---|
| MNIST | MLP | Reference | 0.81 | $5.40\times10^{-2}$ | 0.84 | 1.19% | 0.8534 |
| | | Par. Train | 0.80 | $5.78\times10^{-2}$ | 0.83 | 1.18% | 0.8542 |
| | | Adv. Train I | 1.17 | $9.46\times10^{-2}$ | 1.17 | 4.11% | 0.8280 |
| | | Deep Defense | **1.64** | $\mathbf{1.53\times10^{-1}}$ | **1.58** | **33.18%** | **0.8181** |
| | LeNet | Reference | 1.48 | $1.29\times10^{-1}$ | 1.40 | 26.17% | 0.9074 |
| | | Par. Train | 1.50 | $1.50\times10^{-1}$ | 1.58 | 23.06% | 0.8981 |
| | | Adv. Train I | 1.90 | $2.05\times10^{-1}$ | 1.71 | 50.67% | 0.8810 |
| | | Deep Defense | **2.05** | $\mathbf{2.36\times10^{-1}}$ | **1.84** | **64.54%** | **0.8760** |
| CIFAR-10 | ConvNet | Reference | 0.18 | $5.27\times10^{-3}$ | 0.29 | 21.34% | - |
| | | Par. Train | 0.24 | $8.02\times10^{-3}$ | 0.33 | 27.91% | - |
| | | Adv. Train I | 0.21 | $6.37\times10^{-3}$ | 0.31 | 27.08% | - |
| | | Deep Defense | **0.36** | $\mathbf{1.58\times10^{-2}}$ | **0.47** | **45.05%** | - |
| | NIN | Reference | 0.30 | $1.05\times10^{-2}$ | 0.41 | 34.41% | - |
| | | Par. Train | 0.31 | $1.07\times10^{-2}$ | 0.41 | 36.59% | - |
| | | Adv. Train I | 0.37 | $1.76\times10^{-2}$ | 0.48 | 45.51% | - |
| | | Deep Defense | **0.40** | $\mathbf{2.15\times10^{-2}}$ | **0.50** | **51.07%** | - |
| ImageNet | AlexNet | Reference | 0.29 | $5.46\times10^{-4}$ | - | - | - |
| | | Deep Defense | **0.45** | $\mathbf{8.70\times10^{-4}}$ | - | - | - |
| | ResNet | Reference | 0.69 | $6.96\times10^{-4}$ | - | - | - |
| | | Deep Defense | **1.03** | $\mathbf{1.08\times10^{-3}}$ | - | - | - |

In this paper, we leave the optimal choice of evaluation metric an open question and simply choose some popular ones following previous works. Here in the supplementary material we try to test as many as possible to verify the effectiveness of our method extensively.

In the main body of our paper, we utilize the normalized $l_2$ norm of required adversarial perturbations to evaluate the robustness of different models, as suggested in the DeepFool paper [6]. We notice that in some papers, an unnormalized norm is used instead, which means

$$l_2 := \frac{1}{|\mathcal{D}|} \sum_{k\in\mathcal{D}} \|\Delta_{\mathbf{x}_k}\|_2 \qquad (1)$$

Figure 5: Comparison of different defense methods under the FGS attack. For each network, we report the success rate of FGS with varying $\epsilon$. **Lower is better**. Best viewed in color.

can also be calculated as a metric (see the fourth column of Table 2). In addition, we further evaluate the robustness of different models under the C&W's $l_2$ attack [1], using the official CleverHans [7] implementation. The (unnormalized) $l_2$ values under the C&W's $l_2$ attack are reported in the sixth column of Table 2. Using different reference models trained with different initializations lead to very similar results in our experiments, so we simply omit such variance (e.g., for $l_2$, it is less than 0.003).

Also, when the FGS attack is adopted, the robustness can be evaluated by replacing the $l_2$ norm with an $l_\infty$ norm in the definition of $\rho$ as the FGS attack is usually considered as an $l_\infty$ norm-based (or max-norm based) perturbation method, and get

$$\rho_\infty := \frac{1}{|\mathcal{D}|} \sum_{k \in \mathcal{D}} \frac{\|\Delta_{\mathbf{x}_k}\|_\infty}{\|\mathbf{x}_k\|_\infty}. \tag{2}$$

in the fifth column of Table 2. Higher $l_2$ and $\rho_\infty$ indicate better robustness to the $l_2$ and $l_\infty$ attacks, respectively. Recall that, to establish a benchmark, we adjust $\epsilon$ such that 50% of the image samples are misclassified by well-trained models, as introduced in the main body of our paper. Here we further compare the FGS success rates with respect to varying $\epsilon$ on different models in Figure 5.

As an additional $l_\infty$ attack, the PGD-based method [4] is also tested here. We set $\epsilon = 0.1$ for MNIST, $\epsilon = 0.01$ for CIFAR-10, and compare prediction accuracies on adversarial examples in the seventh column of Table 2. It can be seen that the superiority of our method holds on various baseline networks. Recently, Rozsa et.al. [8] propose a psychometric perceptual adversarial similarity score named *PASS*, which seems consistent with human perception. The lower such score is, the better defensive performance the model gets. We calculate it using an official implementation provided by the authors and report some results in the last column of Table 2.

## 2  Comparison with Adv. Train II

As introduced in the main body of our paper, various forms of adversarial training have been adopted in previous works [10, 2, 6, 3, 5, 4]. Here we test Goodfellow et al.'s adversarial training (abbreviated as "Adv. Train II"). In addition, we also try combining it with our Deep Defense by simply adding the cross entropy loss corresponding to FGS adversarial examples to the training objective of our method. The performance of Adv. Train II, our Deep Defense and our combined method are compared in Figure 6. For each network, we report the $\rho_2$ values under DeepFool in the left column and success rate of FGS with varying $\epsilon$ in the right column.

For our Deep Defense, we fix $\lambda$ and $d$ and vary only $c$ in the figure, while for the combined method, we further fix $c$ and vary only $\epsilon$, as for Adv. Train II. In the right column, we select winning Adv.

Figure 6: Comparison with Adv. Train II on both MNIST and CIFAR-10 datasets. For each network, we report the $\rho_2$ values with DeepFool in the left column (**upper right is better**) and the success rate of FGS with varying $\epsilon$ in the right column (**lower is better**). Best viewed in color.

Train II models (under the FGS attack) from those tested in the left. Obviously, we see that our Deep Defense outperforms Adv. Train II as well in most cases. Moreover, by combining them, we gain even better robustness and benign-set accuracies, which verifies our previous claim of orthogonality.

# 3 MNIST Visualization Results

Quantitative results in our paper demonstrate that an adversary has to generate larger perturbations to successfully attack our regularized models. Intuitively, this implies that the required perturbations should be perceptually more obvious. Here we provide visualization results in Figure 7. Given a clean image from the test set (as illustrated in Figure 7(a)), the generated DeepFool adversarial examples for successfully fooling different models are shown in Figure 7(b)-7(e). Obviously, our method yields more robust models in comparison with the others, by making the adversarial examples closely resembling real "8" and "6" images. More interestingly, our regularized LeNet model predicts all examples in Figure 7(a)-7(d) correctly as "0". For the lower adversarial example in Figure 7(e), it makes the correct prediction "0" with a probability of 0.30 and the incorrect one (i.e., "6") with a probability of 0.69.

Figure 7: An image ($\mathbf{x}_k$) labelled "0" from the MNIST test set with DeepFool examples generated to fool different models including: (b) the references, (c)-(e): fine-tuned models with Adv. Train I, Parseval training and our Deep Defense method. Arrows above the pictures indicate which class the examples are "misclassified" to and the numbers below indicate values of $\|\Delta_{\mathbf{x}_k}\|_2/\|\mathbf{x}_k\|_2$. Upper images are generated for MLP models and lower images are generated for LeNet models.

# 4 CIFAR-10 Convergence Curves

Convergence curves on CIFAR-10 of different methods are provided in Figure 8.

Figure 8: Convergence curves on CIFAR-10: (a)-(b) test accuracy and $\rho_2$ of ConvNet, and (c)-(d) test accuracy and $\rho_2$ of NIN. "Clean" indicates fine-tuning on benign examples. Best viewed in color.

# 5 ImageNet Results

Our method yields models with substantially improved robustness and no accuracy loss is observed on benign test sets, even on ImageNet. Though also enhance models, Parseval and adversarial training

seem difficult to achieve good trade-offs between robustness and accuracy in our experiments on ImageNet. On AlexNet, we were unable to find a suitable $\beta$ such that the fine-tuned model shows reasonably high accuracy ($> 56\%$) and significantly improved robustness simultaneously for Parseval training. This phenomenon can possibly be caused by insufficient hyper-parameter search. For Adv. Train I and II, we observed a decrease of inference accuracy on benign examples when the fine-tuning process starts, and after 10 epochs the accuracy is still unsatisfactory. However, Kurakin et.al. [3] have produced an Inception v3 model [9] using 50 machines after $150k$ iterations (i.e.roughly 187 epochs) of training and obtain only slightly degraded accuracy, so we guess more training epochs and sophisticated mixture of clean and adversarial examples are required.

# 6 Network Architectures and Hyper-parameters

Some hyper-parameters for our fine-tuning are summarized in Table 3. Other hyper-parameters like momentum and weight decay are kept as the same as training the reference models (i.e., momentum: 0.9, and weight decay: 0.0005). Table 4 shows the architecture of networks used in our MNIST and CIFAR-10 experiments. For AlexNet and ResNet experiments, we directly use the reference models from the Caffe and PyTorch model zoos.

Table 3: Some hyper-parameters in the fine-tuning process.

| Dataset | Batch Size | #Epoch | Base Learning Rate |
|---|---|---|---|
| MNIST | 100 | 5 | $5\times10^{-4}$ |
| CIFAR-10 | 100 | 50 | $5\times10^{-4}$ |
| ImageNet | 256 | 10 | $1\times10^{-4}$ |

Table 4: Network architectures adopted in MNIST and CIFAR-10 experiments. We use Conv-[kernel width]-[output channel number], FC-[output channel number], MaxPool-[kernel width], AvgPool-[kernel width] to denote parameters of convolutional layers, fully-connected layers, max pooling layers and average pooling layers, respectively.

| MNIST | | CIFAR-10 | |
|---|---|---|---|
| MLP | LeNet | ConvNet | NIN |
| Input (28×28) | | Input (32×32) | |
| FC-500 | Conv-5-20 | Conv-5-32 | Conv-5-192 |
| ReLU | MaxPool-2 | MaxPool-2 | ReLU |
| FC-150 | Conv-5-50 | ReLU | Conv-1-160 |
| ReLU | MaxPool-2 | Conv-5-32 | ReLU |
| FC-10 | FC-500 | ReLU | Conv-1-96 |
| | ReLU | AvgPool-2 | ReLU |
| | FC-10 | Conv-5-64 | MaxPool-2 |
| | | ReLU | Conv-5-192 |
| | | AvgPool-2 | ReLU |
| | | Conv-4-64 | Conv-1-192 |
| | | ReLU | ReLU |
| | | Conv-1-10 | Conv-1-192 |
| | | | ReLU |
| | | | AvgPool-2 |
| | | | Conv-3-192 |
| | | | ReLU |
| | | | Conv-1-192 |
| | | | ReLU |
| | | | Conv-1-10 |
| | | | AvgPool-8 |

# 7 Note on Max-unpooling Layers of the Reverse Network

In the main body of our paper, we mimic the DeepFool attack calculation using a neural network. In order to do this, the forward process of the "reverse" network should generate an exact output as the backward process of the original classification network. As discussed in the main paper, feasible "reverse" layers can always be made available when building the reverse network.

Special attention should be paid when reversing max-pooling layers. In many modern DL frameworks (including TF, PyTorch and Keras), the forward process of a max-unpooling layer is not strictly equal to the backward process of a max pooling layer, if the stride is smaller than the pooling window size. In the max pooling operation, if more than one overlapped sliding windows select the same element from feature maps simultaneously, the derivatives from later feature maps should be summed up in the backward calculation. However, many DL frameworks just select one of the overlapped windows and ignore the others in the forward process of a max unpooling, which is slightly different. Such difference could accumulate layer-by-layer and the final perturbation can be very different from the original DeepFool, especially for deep networks. We release a patch to fix this along with our source code at `https://github.com/ZiangYan/deepdefense.pytorch`.

## Footnotes

*The first two authors contributed equally to this work.