[Reviews · NeurIPS 2018]

Reviewer 1



The paper proposes Deep Defense, an algorithm for training deep NN to be more resistant to adversarial test samples. In order to defend against attacks that use the gradient of the classification function (f) the central idea is to add a regularization term into the training objective that prefers functions f that are resistant to (reasonably small magnitude) gradient attacks. Thus the aim is to directly identify classification functions f that are more resistant the types of attacks generated by Deep Fool and other similar algorithms. The idea is not entirely novel and there is already a class of methods based on regularization. However in its implementation, the specifics of the proposed method seem to be novel. The authors claim that the main benefit is that they do not have to employ approximations or optimize potentially loose bounds and may thus defend against these attacks more successfully (PS: While I publish papers on Deep NN, I'm not an expert on adversarial attacks and would defer to other reviewers who are more familiar with the literature regarding novelty). Experimental comparisons are not very rigorous. For example no error bars are provided so it is hard to even guess whether superiority of the method is statistically significant or an artifact of random chance. However, it seems to be slightly more robust compared to state of the art methods from recent papers including Deep Fool and Parseval training. I found the paper to be not entirely clear or self-contained in explaining its concepts. For the most part this is entirely avoidable and is a result of not making enough of an effort to make the paper more accessible. For example, why not just provide equation 6 and plug in the exponential function right from eqn 3? Why keep us guessing about the form of the R till the end, needlessly confusing the issue? This would avoid lots of opaque discussion and would make it clear from the beginning what they are proposing. Such simple means would make the paper much more readable and easily reproducible especially since the concepts are relatively straightforward when examined from first principles (similarly lines 126 and 127 should just explain the concept instead of referring the reader to [26]).

Reviewer 2



The main idea of this paper is contribution of training recipe named "deep defense" that uses modification of classification loss with perturbation based regularizer and uses the encoder/decoder based architecture to achieve results on MNIST, CIFAR-10 and Image Net datasets that surpass ones from chosen reference implementation, DeepFool and FGS adversarial implementations and shows robustness that surpasses ones from above adversarial algorithms as well, by a wide margin. The paper builds on the perturbation based regularization method for adversarial training with non-equal (weighted) treatment of robust examples (versus the others). The algorithm is described well, though sometimes than referring to the reference for the important details, would be good to include them, esp. in context of the setup, formulae and notations. The authors do a good job of supporting their reasoning for the choice of the perturbation regularization term, that has the advantage of utilizing the network architecture without any changes, and can be utilized in encoder/decoder framework, with initializations from pre-trained networks to achieve robustness to adversarial attacks while maintaining high accuracy on benign examples. The claims from the authors are well supported by the results in Table 1 and 2 and the Figures 2-4 that show that Deep Defense beats the other two adversarial methods (Deepfool and FGS) methods on MNIST, CIFAR-10 datasets and is more robust across the board and also surpasses the reference network itself in benign accuracy including the ImageNet. They also demonstrate the impact of changing the hyperparameters c and lambda and layer regularization and how the accuracy varies with epochs. Authors intend to open source implementation. A few areas of explanations and improvements that would help understanding the paper better are mentioned below - It would be really interesting to add if the authors faced any challenges with deconvolution. In the literature both upsampling/convolution and deconvolution/unpooling have been used in decoders and both have pros/cons, though in the encoder/decoder setup (which the authors for some reason don't call as such), deconvolutions were employed by the authors and would be great to know if there were any challenges (esp. with deeper networks). Also, conspicuously the details on the decoder is missing from the description in the supplement. - details on R the exponential function in the perturbation regularization term aren't provided (its just mentioned as such) - the paper mentions all operations in CNN being differentiable and mention ReLU, that sounds off-point - What is the stopping criteria (different epochs/batches/iterations are mentioned for different datasets but it wasn't clear - In Table 1 the Perturbed examples from DeepFool were chosen when the Deep Defense drops to 50%, would good to know a) what if the perturbed data was instead chosen from one of the other two adversarial algorithms (so for e.g. when their performance drops to 50%) b) what happens below 50% was chosen to start with - For Image Net the authors conclude insufficient hyper-parameter exploration to include results for the two adversarial algorithms, would be good to include that in final version - The choice of the networks, why Alexnet and Resnet-18 were chosen (the authors make a remark about deep neworks, and would like to get their input on deeper architectures)

Reviewer 3



Summary of the paper: This paper presents a training procedure to increase a network's robustness to adversarial attacks. The main idea is that the normal cross-entropy loss function is augmented with another loss term penalizing low norms of the adversarial perturbation (such as computed by "DeepFool" or the "fast gradient sign" method) for the training data. A slightly more sophisticated version of the adversarial loss is actually used where for correctly classified training examples low norms of the perturbation are penalized while for incorrectly classified training examples high norms of the perturbation are penalized. + Convincing results are obtained. The training regime presented produces a better defence to FGS adversarial examples on MNIST and CIFAR-10 datasets then other defensive schemes such as Parseval networks, extending the training set with DeepFool adversarial examples. The regime also increases the Though the boost becomes less pronounced for the more sophisticated networks. The introduced training regime also gives a slighter bigger bump in test accuracy on the benign set (when updating a pre-trained model using fine-tuning with the augmented loss function) than the those achieved by the other training regimes. + The paper presents a straight-forward and sensible idea to help make a network more robust to common adversarial attacks while at the same time not affecting accuracy on the benign set. - The details of how the "adversarial" loss is implemented are a bit unclear to me from reading section 3.3 and examining figure 1. Is the exact DeepFool procedure for constructing an adversarial perturbation replicated or is it approximated? The authors say they build a recursive regularization network which, I think, emulates calculating one update step in the DeepFool method for computing the adversarial perturbation. Then is this network recursively applied until an adversarial example is found for each training example at each update step? - There are missing details on the effect on the new loss on the practicalities of training. How much slower does the introduced training regime make one epoch of training? On average how many updates are required to generate an adversarial perturbation for a training example? How is the convergence of the training effected etc? - The practical details of some aspects of training are missing some details. When computing the adversarial loss for training example how is the target class chosen for the adversarial example chosen? And does this approach differ for the correctly and the incorrectly classified training examples... Without these details it would be hard to reproduce the results. A summary of my rationale for the paper is that the idea in the paper is nice (not revolutionary but nice) and produces good results. I have some problems with the presentation and clarity of the technical detail. However, I think these could be relatively easily rectified. Comment: I think the authors should include elements of Table 2 in the supplementary material into the main paper. There is a clall in the adversarial community to training regime's robustness to other tougher adversarial attacks than FSG (ie "Bypassing ten detection methods" paper of Carlini and Wagner!) and this table is more in the spirit of this call then the table 1.